# Recent Advances in Potential Health Benefits of Quercetin

**DOI:** 10.3390/ph16071020

**Published:** 2023-07-18

**Authors:** Fatemeh Aghababaei, Milad Hadidi

**Affiliations:** 1Centre d’Innovació, Recerca i Transferència en Tecnologia dels Aliments (CIRTTA), TECNIO-UAB, XIA, Departament de Ciència Animal i dels Aliments, Universitat Autònoma de Barcelona, UAB-Campus, 08193 Bellaterra, Spain; aghababaei.afi@gmail.com; 2Department of Organic Chemistry, Faculty of Chemical Sciences and Technologies, University of Castilla-La Mancha, 13071 Ciudad Real, Spain

**Keywords:** antioxidative activity, free radicals, anticancer activity, anti-inflammatory, chronic diseases, pharmaceutical applications

## Abstract

Quercetin, a flavonoid found in fruits and vegetables, has been a part of human diets for centuries. Its numerous health benefits, including antioxidant, antimicrobial, anti-inflammatory, antiviral, and anticancer properties, have been extensively studied. Its strong antioxidant properties enable it to scavenge free radicals, reduce oxidative stress, and protect against cellular damage. Quercetin’s anti-inflammatory properties involve inhibiting the production of inflammatory cytokines and enzymes, making it a potential therapeutic agent for various inflammatory conditions. It also exhibits anticancer effects by inhibiting cancer cell proliferation and inducing apoptosis. Finally, quercetin has cardiovascular benefits such as lowering blood pressure, reducing cholesterol levels, and improving endothelial function, making it a promising candidate for preventing and treating cardiovascular diseases. This review provides an overview of the chemical structure, biological activities, and bioavailability of quercetin, as well as the different delivery systems available for quercetin. Incorporating quercetin-rich foods into the diet or taking quercetin supplements may be beneficial for maintaining good health and preventing chronic diseases. As research progresses, the future perspectives of quercetin appear promising, with potential applications in nutraceuticals, pharmaceuticals, and functional foods to promote overall well-being and disease prevention. However, further studies are needed to elucidate its mechanisms of action, optimize its bioavailability, and assess its long-term safety for widespread utilization.

## 1. Introduction

In recent times, there has been a significant increase in the utilization of natural bioactives for the management of chronic ailments owing to their low toxicity and environmentally friendly characteristics. Quercetin, a bioactive molecule, has been the subject of extensive research owing to its diverse pharmacological activities, including antioxidant, antiviral, immune-modulatory, and anticancer activity with a low toxicity profile. Despite numerous reports suggesting the therapeutic efficacy of quercetin, it is yet to be recognized as an established molecule in the pharmaceutical industry due to the lack of clinical data and an exact understanding of its mechanism of action [1]. Additionally, a great deal of chemical compounds and phytoconstituents with a wide range of pharmacological and biological activities have been extracted and discovered from medicinal herbs [2,3].

Quercetin (3,5,7,3′,4′-pentahydroxyflavone) is widely distributed in flowers, bark, stems, roots, wine, vegetables, tea, fruits such as apples, dill, berries, cilantro, lovage, onions, and capers. The most abundant sources of quercetin are shown in Figure 1. It is chemically composed of three benzene rings and five hydroxyl groups.

Quercetin is a crystalline, yellow material which is entirely insoluble in cold water, soluble in alcohol and lipids, and marginally soluble in hot water and has a bitter taste. It is an aglucone or aglycon that does not include any carbohydrate molecules in its structure, and it gives a range of flowers great colors [5]. Free radicals react with other substances rapidly in order to obtain their electrons and become stable. An attack causes a molecule to lose an electron and turn into a free radical, which sets off a series of events that disturb living cells [6]. From the literature, the hydroxyl groups at the A and B rings’ 3, 5, 7, 3′, and 4′, the double bond between the second and third carbons, and the carbonyl group on the fourth carbon were verified to have a major part in the antioxidant capabilities of quercetin [7]. It also exhibits a variety of intriguing treatments for different health issues such as anticancer activity [8], antiallergic activity [9], anti-diabetes activity [10], anti-obesity activity [11], anti-hyperuricemia and gouty arthritis [12], etc. The most important impact of quercetin is its ability to inhibit the spread of certain cancers including those of the breast, cervical, lung, colon, prostate, and liver [13]. In Figure 2, the properties of quercetin and its structure are shown. The capacity to block enzymes that activate carcinogens and diverse processes involving cellular signaling are used to exert the aforementioned anticancer effects.

## 2. Review Methodology

Several specialized electronic databases were searched to identify relevant research on quercetin’s health benefits and biological activities, including ScienceDirect, PubMed/Medline, and Scopus databases. A literature search was conducted using the following MeSH terms: “quercetin/pharmacology”, “anti-cancer/quercetin”, “anti-inflammatory”, “antioxidants/pharmacology”, “antibacterial”, “antifungal”, “antiviral”, “anti-Alzheimer”, “antidiabetic”, “anti-obesity”, “anti-hypertensive”, “anti-asthmatic”, “anti-allergic”, and “bioavailability of quercetin”. There were no limitations on publication type, and all sources used were in English. However, more than 90 percent of the sources used related to articles from the last 5 years.

## 3. Chemical Structure and Main Sources

One of the six subclasses of flavonoid chemicals, known as flavonols, includes quercetin. The name, which has been in use since 1857, is derived from quercetum (oak woodland), in honor of Quercus. It is an inhibitor of polar auxin transport that exists naturally [14]. The scientific name for quercetin according to the International Union of Pure and Applied Chemistry (IUPAC) is 3,3′,4′,5,7-pentahydroxyflvanone or its synonym 3,3′,4′,5,7-pentahydroxy-2-phenylchromen-4-one. This indicates that the OH groups on quercetin are connected at positions 3, 5, 7, 3′, and 4′. The aglycone quercetin (C15H10O7) is devoid of an associated sugar. By substituting a glycosyl group (rhamnose, a sugar like glucose or rutinose) for one of the OH groups (often at position 3), a quercetin glycoside is created. The absorption, solubility, and in vivo effects can all be influenced by the linked glycosyl group. Generally, quercetin glycoside has a higher water solubility than quercetin aglycone because it has a glycosyl group [15]. The connected glycosyl group makes a quercetin glycoside distinct. In general, the term “quercetin” should only be used to refer to the aglycone; but, in research and the supplements sector, the name is occasionally used to refer to quercetin-like compounds, including its glycosides [14].

Quercetin is widely distributed in various plant-based foods. Some of the main plant sources of quercetin include capers, rocket, dill, coriander, fennel, juniper berries, corn poppy, bee pollen, and okra which contain the highest amount of quercetin [4]. The variegated provenances of quercetin demonstrate country-specific and regional discrepancies which can be attributed to the complexity of localized environments, customs, and other factors [16]. In addition, more than twenty plant species, including *Santalum album*, *Mangifera indica*, *Emblica ofcinalis*, *Curcuma domestica valenton*, *Withania somnifera*, *Foeniculum vulgare*, and *Cuscuta refexa* contain quercetin. Due to its extensive range of pharmacological effects, quercetin is frequently utilized as a dietary supplement in capsule and powder form [6]. Depending on consumption of vegetables, tea, and fruits quercetin intake varies from 50 to 800 mg per day; other nations reported varying dietary intakes. Among the investigated foods, it has been discovered that onions contain the most quercetin (approximately 300 mg/kg) [7]. Quercetin and kaempferol were also present in other plants, such as broccoli and kale, albeit in considerably lesser amounts. However, tea has a high catechin content but a low quercetin flavonoid content. As red grapes, cherries, and blueberries have a substantial number of different anthocyanins, the color of fruits and vegetables reflects the amounts of flavonoids present. Additionally, most of these fruits include flavonoids, particularly quercetin. According to reports, the average daily consumption of quercetin in nations like the USA, Spain, Japan, and China is 9.75, 18.48, 16.2, and 18 mg [14].

## 4. Biological Activities

### 4.1. Antioxidant Activity

So far, the antioxidant profile of quercetin is its most well-known characteristic. It shields our body from free radicals and is the most potent natural antioxidant flavonoid [17]. The antioxidant role of hesperidin and quercetin against reducing power assay, DPPH, hydroxyl radicals, nitric oxide, hydrogen peroxide radicals, and superoxide has been demonstrated by Srimathi et al. [18]. Their findings showed that, compared to ascorbic acid, which was used as a reference, both compounds exhibited stronger inhibitory efficacy against DPPH [19]. According to Lesjak et al. [20], by preventing lipid peroxidation, methylation quercetin metabolites (such as isorhamnetin and tamarixetin) showed stronger antioxidant activity than quercetin. As a scavenger of reactive oxygen species, quercetin has been demonstrated to possess antioxidant qualities [21].

Hydrogen peroxide is a key reactive oxygen species in many neurological disorders [22]. In past research, human neuronal SH-SY5Y cells were used for examining the neuroprotective properties of quercetin against H_2_O_2_-activated apoptosis. The presence of quercetin was found to reduce lactate dehydrogenase and H_2_O_2_-mediated cytotoxicity. In addition, quercetin boosted the amount of the antiapoptotic gene Bcl-2 and reduced the level of the pro-apoptotic gene Bax in neural cells. As a result, quercetin prevents neurological diseases brought on by oxidative stress and apoptosis [23]. By controlling antioxidant enzymes and oxidative stress factors, quercetin acts to prevent the progression of a number of cancers, including liver, lung, colon, prostate, cervical, and breast cancers. Moreover, intriguingly, quercetin has been shown to be an effective antioxidant in reducing and inhibiting damage and oxidative stress in both in vitro and in vivo studies. Quercetin has been demonstrated to have antioxidant and hepatoprotective effects against acute liver damage brought on by tertiary butyl hydrogen peroxide in in vivo investigations. By removing free radicals and raising the amounts of endogenous antioxidants, quercetin efficiently shields cells against radiation-induced damage and genetic toxicity [24]. As part of an in vivo study, rats were measured for histology and markers of oxidative stress, including lipid peroxidation (LPO) and hydrogen peroxide (H_2_O_2_), in order to evaluate the antioxidant activity of quercetin when compared to testosterone and a carcinogen. Researchers discovered that, compared to rats treated with quercetin, animals treated with testosterone and carcinogens had lower levels of GSH and greater amounts of H_2_O_2_ and LPO [25]. In a different study, this author noted that quercetin raised the concentrations of antioxidant enzymes and apoptotic proteins in rats with prostate cancer. Additionally, they discovered that quercetin controlled the expression of proteins that are elevated in cancer, including insulin-like growth factor receptor 1 (IGFIR), protein kinase B (AKT), androgen receptors (AR), and cell proliferation and anti-apoptotic proteins [26]. Due to its antioxidant qualities and inhibitory action on the ROS/NF-B/NLRP3 inflammasome/IL-1/IL-18 pathways through the generation of HO1, quercetin has beneficial benefits in the treatment of alcoholic hepatitis. In the meanwhile, quercetin also boosts IL-10, a substance that reduces inflammation, independently of HO1. To prevent liver damage from acute alcohol intoxication, quercetin inhibits the release of inflammatory factors and activation of the NLRP3 inflammasome while increasing the production of IL-10 and HO-1 [27]. Nevertheless, in several experiments, quercetin prevented oxidative stress and neurodegenerative disorders brought on by cadmium fluoride as well as radiation-induced brain damage in rats, acrylamide-induced oxidative stress in rats, radiation-induced brain damage in rats, and nerve damage in diabetic rats’ retinas [28,29]. Table 1 gives a summary of the antioxidant activity of quercetin.

### 4.2. Anticancer Activity

Quercetin has been shown to have substantial anticancer action (Figure 3). Numerous studies have reported that quercetin can induce apoptosis and cell cycle arrest in different cancer cell lines such as those of breast, prostate, lung, and colon cancers. According to the study, administering nano-quercetin to MCF-7 breast cancer cells increases mRNA expression and apoptosis [49]. Additionally, it was shown that quercetin decreased the expression of genes multidrug resistance protein 1 and NAD(P)H quinone oxidoreductase 1 and sensitized MCF-7 cells to the chemotherapy medication doxorubicin (Dox) [49,50]. According to a study, quercetin inhibited cell viability in colon 38 (MC38) and colon 26 (CT26) cells, induced apoptosis in CT26 cells through the MAPK pathway, inhibited cell viability through the MAPK pathway, inhibited cell viability through the MAPK pathway in CT26 cells, and regulated the expression of EMT markers by nontoxic doses of quercetin. Inhibiting CT26 cells’ migration and invasion abilities by inhibiting their expression of tissue inhibitors of metalloproteinases (TIMPs) inhibits their invasion and migration abilities [51].

The limited solubility of quercetin in medicine is one of its drawbacks. Guan et al. [52] created quercetin (QT)-loaded PLGA-TPGS nanoparticles (QPTN) and tested their therapeutic effectiveness for liver cancer as a means of overcoming these drawbacks. Results showed that QPTN could inhibit tumor development by 59.07% and could cause HepG2 cell death in a concentration-dependent manner. These researchers concluded that due to the improved pharmacological impacts of quercetin with higher liver targeting, QPTN could possibly be employed in a viable intravenous dose form for liver cancer treatment [52]. The effectiveness of androgen-independent prostate cancer cells was examined by many research teams in relation to natural products like quercetin. Authors discovered that quercetin achieves the following: enhances the healing practicality of Doc; significantly slows tumor progression; reduces Ki67; raises cleavage of caspase 7; decreases blood concentrations of growth factors like epidermal growth and VEGF factor; and significantly raises the levels of tumor silencers mir15a and mir330 [53,54]. Additionally, quercetin has anticancer effects in xenograft mice implanted with human gastric carcinomas caused by the Epstein–Barr virus ((EBV)(+) or EBV(−)). In tumor tissues from mice injected with EBV(+) human gastric carcinoma, these scientists discovered that quercetin has anticancer impact in these cells by impeding EBV viral protein production like that of EBNA1 and LMP2 proteins. In EBV(+) human gastric cancer, quercetin more effectively induced p53-dependent apoptosis than isoliquiritigenin, and this result was correlated with increased expression of caspases 3, 9, and Parp. Quercetin stimulated the expression of p53, Bax, and Puma as well as caspases 3, 9, and Parp at comparable amounts in EBV(−) human gastric carcinoma (MKN74) [55,56]. A summary of the anticancer activities of quercetin are shown in Table 2.

### 4.3. Antibacterial Activity

Quercetin has demonstrated strong bacteriostatic action against a variety of bacterial species, including *Helicobacter pylori*, *Salmonella enterica* serotype Typhimurium, *Micrococcus luteus*, *Yersinia enterocolitica, Pseudomonas aeruginosa*, *Escherichia coli*, *S. aureus*, *P. fluorescens, Campylobacter jejuni*, and *Staphylococcus epidermidis*, that has proved more efficient against Gram-positive compared to Gram-negative bacteria [78]. Quercetin derived from *C. ofcinalis* was shown to have antibacterial properties. Different organic extracts were used in in vitro tests against *S. aureus*, and an in silico analysis was performed on four key bacterial enzymes, specifically utilizing AutoDock Vina 1, DNA primases, undecaprenyl pyrophosphate synthase, peptide deformylase, and gama hemolysins. According to the findings of an in vitro investigation, quercetin at a level of 20 mg/mL totally inhibits the development of *S. aureus*. Quercetin was discovered to be the secondary metabolite that *S. aureus* used to obstruct three of its proteins, based on in silico studies [6]. A study conducted by Osonga et al. [79] indicated that quercetin derivatives (including quercetin 4′,5-diphosphate, quercetin 3′,4′,3,5,7-pentaphosphate, and quercetin 5′-sulfonic acid) were highly biocompatible, soluble, and highly effective antibacterial agents with 100% inhibition of *Listeria monocytogenes*, *Aeromonas hydrophila*, and *Pseudomonas aeruginosa*. The antibacterial profile of 13 flavonoids, including quercetin, luteolin, apigenin, rutin, isoorientin, chrysin, kaempferol, flavones, orientin, isovitexin, vitexin, naringin, and vitexin 2-o-rhamnoside, along with 6 organic acids, namely citric acid, chlorogenic acid, quinic acid, rosmarinic acid, malic acid, and salicylic acid, has been reported. Four clinical isolates, namely Gram-positive bacteria *E. faecalis* and *S. aureus* as well as Gram-negative *E. coli* and *P. aeruginosa*, were used to estimate the MIC values of all the test compounds using the microdilution technique. It was determined that all tested substances provided antibacterial effects against Gram-negative bacteria to a greater extent in comparison to Gram-positive bacteria. A further finding of the study was that the hydroxyl groups in phenyl rings had no effect whatsoever on the level of activity. The hydroxy derivatives of flavone, however, showed a significant increase in activity against *S. aureus* [80].

A new complex with nanoparticles based on chitosan made of quercetin and catechin was described in [81]. Chitosan and sodium tripolyphosphate were combined with genipin to create this complex using the ionic gelation process. The aim was to improve catechin and quercetin’s antioxidant and antibacterial profiles. The measured zeta potential was 31.79 1.28 mV, and the average particle size was 180.4 nm Nanoparticles in vitro demonstrated sustained drug release. Three bacterial strains, namely *B. subtilis*, *E. coli*, and *S. aureus*, were used in an antibacterial experiment (zone of inhibition and lowest inhibitory concentration) of pure chemicals, blank nanoparticles, and drug-loaded nanoparticles. The findings supported the increased antibacterial profile of chitosan-based nanoparticles (G-C-Q NPs) rich in catechin and quercetin. Therefore, polymeric particles are a promising option for bacterial infections as a medication delivery mechanism [81]. Table 3 gives a summary of the antibacterial, antifungal, anti-inflammatory, antiviral, and antidiabetic activities of quercetin.

### 4.4. Antifungal Activity

Quercetin exhibits antifungal activity against various pathogenic fungi. It has been reported to inhibit the growth of Candida species, *Cryptococcus neoformans*, Aspergillus species, and dermatophytes, among others. Quercetin inhibits fungal growth by inducing oxidative stress and altering the cell membrane composition of fungi, leading to cell death. Additionally, quercetin has been found to synergize with other antifungal agents, making it a promising alternative or adjunctive therapy in the treatment of fungal infections. According to Rocha et al. [82], *C. parapsilosis*, *C. metapsilosis*, and *C. orthopsilosis* strains were tested for kaempferol’s and quercetin’s minimal inhibitory concentrations (MIC). Kaempferol and quercetin were found to reduce the biomass and metabolic activity of all the studied strains. While kaempferol’s MIC varied from 32–128 ug/mL, quercetin’s was between 0.5 and 16 ug/mL. These substances can, therefore, be employed as antifungals. Additionally, quercetin had the highest antifungal effects against *Cryptococcus neoformans*, *Aspergillus niger*, and *Candida albicans* [5]. In another study, quercetin, an aqueous sodium citrate, and a trichloro gold hydrochloric acid solution were used to create quercetin-capped gold nanoparticles (AuNPsQct). By using TEM, FTIR, and UV analyses, nanoparticles were confirmed. Results demonstrate that AuNPsQct have potent antifungal activity against Aspergillus fumigatus at doses between 0.1 and 0.5 mg/mL [83]. The survival strategies of *Aspergillus flavus* throughout the germination phase were described in [84]. Quercetin’s impact on *A. flavus* signaling pathways during germination was examined in this study utilizing scanning electron microscopy (SEM). Real-time quantitative PCR was used to detect a substantial increase in heat shock proteins and calcium signaling pathways in response to quercetin at 4- and 7-h intervals. In groups given quercetin, higher concentrations of the genes encoding Rhogdp, calcium kinase, cAMP, Pkc, and Plc were found. Thus, the evidence pointed to quercetin’s potential as an antifungal [84]. When used alone, no antifungal impact of quercetin was reported on *Clostridium neospora*, but when combined with AmB (amphotericin B), the antifungal activity was significantly enhanced. This suggests that quercetin is a possible adjuvant medication for the treatment of AmB caused by fungi [85]. In another interesting study, Gao et al. [86] reported that quercetin is a promising synergistic agent with fluconazole and is an effective antifungal medication in the clinical care of *Candida vaginitis* brought on by *Candida albicans* biofilms.

### 4.5. Anti-Inflammatory

It has been demonstrated that quercetin has strong anti-inflammatory effects and a lengthy half-life. Quercetin and galangin, two flavanols, were shown to have anti-inflammatory properties in research utilizing lipopolysaccharide-stimulated RAW264.7 macrophages. A mouse model of 2,4-dinitrochlorobenzene-induced atopic dermatitis was employed. For examining the synergistic impact, both drugs were administered both singly and in combination. The study’s findings showed that both of these substances lower levels of interleukin-6, NF-kB, and nitric oxide production. Based on histological analysis and measurements of ear thickness, it was shown that using these flavonoids in combination significantly lowered IgE levels and inflammation. As a result, the combination of quercetin and galangin offered novel AD-preventive strategies [87]. Quercetin inhibits the growth of IL-8-induced LPS in lung A549 cells and the creation of lipopolysaccharide (LPS)-mediated tumor necrosis factor (TNF-α) in macrophages, according to a number of in vitro studies [14]. Additionally, quercetin can diminish quantities of TNF-α and interleukin (IL)-1α produced by LPS, which lessens the amount of apoptotic neuronal cell death driven by activated microglia [88]. Inflammatory pneumonia and severe inflammation are risk factors for SARS-CoV-2. Acute lung damage, multiple organ dysfunction syndromes, and acute respiratory distress syndromes are the leading causes of mortality in infected individuals owing to cytokine storm. According to a study, NLRP3 is an inflammasome that activates multiple inflammatory mediators, including NRF2, TXNIP, and SIRT1. Quercetin may be used as a therapy for intense inflammation and in situations where life is at risk, such as COVID-19, since it successfully inhibited NLRP3 by acting on this inflammasome [89]. In the normal condition of peripheral blood mononuclear cells (PBMC), quercetin significantly upregulates the gene expression and production of interferon-γ (IFN-γ), which is obtained from T helper cell 1 (Th1), and downregulates IL-4, which is obtained from Th2. Additionally, quercetin is known to decrease the production of inflammatory molecules COX-2, nuclear factor-kappa B (NF-κB), activator protein 1 (AP-1), mitogen-activated protein kinase (MAPK), reactive nitric oxide synthase (NOS), and reactive C-protein (CRP) [90]. Pro-inflammatory stimuli and cytokines elevate blood glucose levels and have an impact on endothelial cell activity. It was shown that quercetin can mitigate the negative effects of inflammation and hyperglycemia on vascular endothelium. A total of 27 metabolites produced by human umbilical vein endothelial cells (HUVEC) were identified and measured using the metabolomics method. After treating HUVECs with high glucose concentrations, a significant rise in lactate and glutamate levels was detected. Quercetin decreased adenosine triphosphate and lactate concentrations while elevating inosine levels. Quercetin and TNF-γ decreased levels of pyruvate. Both on its own and in the presence of pro-inflammatory stimuli, quercetin changed the balance of HUVEC metabolites toward a less inflamed phenotype. Quercetin suppressed the production of inflammatory cytokines such as IL-6, TNF-α, and IL-1β via upregulating TLR4. Subsequently, repressed receptors yielded targeted suppression of IκBα phosphorylation and MAPK expression and thereby inhibited the nuclear transfer of NF-κB. The activated PPAR-γ also restrained inflammatory mediators through NF-κB. Quercetin employed suppression on the phosphorylation of PKCδ to control the PKCδ–JNK1/2–c-Jun pathway. This pathway arrested the accumulation of AP-1 transcription factor in the target genes, thereby resulting in reduced ICAM-1 and inflammatory inhabitation. Quercetin overexpressed Nrf2 and targeted its downstream gene, contributing to increased HO-1 levels responsible for the down-regulation of TNF-α, iNOS, and IL-6 (Figure 4) [91]. Moreover, it has been found that topical inflammation is unaffected by quercetin because of its poor absorption through the skin’s surface, but pentamethyl ether, a quercetin derivative, has demonstrated strong anti-inflammatory properties with extreme absorption through the skin’s surface in rats [92].

**Table 3 pharmaceuticals-16-01020-t003:** Other biological properties of quercetin.

Therapeutic Agent	Biological Activity	Activity Performed	Key Findings	References
Quercetin	Antibacterial	*S. aureus*	Prevented the growth of *S. aureus*	[93]
*Aspergillus niger*’s transformation of rutin to quercetin	Antibacterial	*S. aureus*, *E. coli*, and *P. aeruginosa*	Increased activity against *S. aureus* was reported	[94]
6 organic acids + 13 flavonoids	Antibacterial	*E. faecalis*, *S. aureus*, *E. coli*, and *P. aeruginosa*	All compounds showed activity against Gram-negative bacteria	[80]
Quercetin	Antibacterial	*S. saprophyticus* and resistant *S. aureus*	Antibacterial effects against MRSA, MSSA, VRSA, and VISA	[95]
Quercetin with kaempferol	Antifungal	*C. metapsilosis, C. orthopsilosis*, and *C. parapsilosis*	Quercetin was more effective than kaempferol as an antifungal agent	[82]
Quercetin/rutin and amphotericin B	Antifungal	*Candida species* and *Cryptococcus neoformans*	Evidence of synergistic antifungal action	[85]
Quercetin with fluconazole	Antifungal	Vulvovaginal candidiasis	Quercetin and fluconazole have a synergistic antifungal action	[86]
Quercetin	Antifungal	*Aspergillus flavus*	Against *Aspergillus flavus*, quercetin exhibited antifungal properties	[84]
Quercetin and galangin	Anti-inflammatory	Atopic dermatitis in rats	Reduced inflammation due to decreases in NF-kB, interlukin-6, and nitric oxide	[87]
Quercetin	Anti-inflammatory	Endothelial cell function	Alteration of HUVAC in order to prevent inflammation	[96]
Quercetin	Anti-inflammatory	Inflammasome NLRP3	NLRP3 suppression results in a decrease in inflammation	[89]
Quercetin	Antiviral	HCV	Decreased viral load	[97]
Quercetin glucoside	Antiviral	Zika virus	Quercetin glucoside’s cytotoxic effects on the Zika virus	[98]
Quercetin	Anti-diabetes	Diabetes-induced osteopenia	Normal bone structure and blood sugar levels in the group treated with quercetin	[99]
Quercetin SEDDS	Anti-diabetes	Diabetes-induced streptozotocin	Antihyperglycemic effects of quercetin increased	[100]
Quercetin-loaded Soluplus micelles	Anti-diabetes	Rat in vivo model	Reduction in blood glucose levels due to the increased bioavailability of quercetin	[101]

### 4.6. Anti-Alzheimer Activity

Alzheimer’s disease (AD) is a chronic NDD that causes severe atrophy of the cingulate gyrus, frontal cortex, parietal lobes, and temporal due to the loss of neurons and synapses in the cerebral cortex and some subcortical regions [102]. Memory loss and mental impairments, such as agnosia, aphasia, and apraxia, are characteristics of AD, a chronic neurodegenerative situation that is reported to be the most common cause of dementia connected to neuroinflammatory processes in the central nervous system [103]. The potential health benefits of quercetin against AD due to inhibition of acetylcholinesterase have been demonstrated [5]. One key mechanism is the reduction of oxidative stress in the brain, which is caused by an imbalance between reactive oxygen species (ROS) and antioxidant defense mechanisms. Quercetin acts as a potent antioxidant by scavenging ROS, inhibiting lipid peroxidation, and enhancing the activity of antioxidant enzymes. This helps to counteract oxidative stress and protect against neurodegenerative processes that contribute to AD. Another mechanism is the modulation of neuroinflammation, which is a hallmark of AD. Quercetin inhibits the activation of microglial cells and reduces the production of pro-inflammatory cytokines, thereby mitigating the detrimental effects of chronic inflammation in the brain. Overall, quercetin may exert its anti-Alzheimer effects by reducing oxidative stress and modulating neuroinflammation [33,104]. There are several forms of memory, including verbal, olfactory, episodic, and visual. These are divided into two groups: implicit (nonverbal habitual memory) and explicit (active or passive recollection of facts) [105]. Considering the fact that aged neurons, astrocytes, and microglia can experience cellular senescence, which is characterized by the release of inflammatory mediators and a decline in the functioning of tissues and organs, it is an essential process in AD [106]. In this respect, it has been demonstrated that quercetin selectively kills senescent cells in the brain of Alzheimer murine models, indicating quercetin has a senolytic effect [106,107].

According to a particular work of research, transgenic mouse lines often had human gene mutations at many locations, including APP, PSEN1, and apoE4. Therefore, not only is the accumulation of plaque simulated but so too is the metabolism of cholesterol and insulin [108]. According to a study conducted by Zhang et al. [109], to determine whether quercetin has any impact on reducing spatial learning and memory impairment, we performed a meta-analysis of the Morris water maze test data from four papers. The cognitive function of quercetin-treated AD models significantly improved, and there was no obvious variability among the independent investigations, according to the statistical findings. The quercetin was used to screen the prospective target genes, and microarray datasets were utilized to find the genes that are differently expressed in AD. Second, identification of the genes targeted by quercetin as AD-related genes was carried out. Finally, using quantitative polymerase chain reaction (qPCR) in HT-22 cells treated with various dosages of quercetin, the anticipated target genes were validated. This study demonstrated a more focused and efficient technique to provide a fresh perspective on quercetin treatment for AD processes [110]. Nevertheless, recent studies have revealed that quercetin increased the pathogenesis of AD and associated cognitive impairments in triple-transgenic, elderly AD mice [111]. In addition, rats treated with chronic rotenone or 3-nitropropionic acid showed enhanced neuroprotection when quercetin and fish oil were taken orally [112]. Quercetin (especially in its methylated form, azaleatin) exhibits a stronger association with the active site of this enzyme than do conventional drugs used in clinical practice, according to in silico analysis, which further demonstrates that it has a superior inhibitory capacity over AChE [113]. In another study, Moonhee Lee et al. [114] compared the neuroprotective effects of quercetin, flavones, chlorogenic acid, and caffeine before concluding that quercetin is the primary component in coffee that has a neuroprotective activity against AD. The neuroprotective effects of quercetin have been attributed to a number of mechanisms, including attenuation of oxidative stress in AD and inhibition of A aggregation, NFT formation, APP-cleaving enzyme (BACE1), acetylcholinesterase (AChE), and others. The anti-Alzheimer activity of quercetin in particular has been associated with a variety of direct and indirect molecular and cellular modes of action. The use of quercetin as a CNS medication that may cross the BBB, however, increases its bioavailability. However, because quercetin is a dietary supplement and nutraceutical, it is not investigated at the molecular and cellular level. Quercetin is a powerful anti-AD lead molecule, and structural changes to the compound may result in a therapeutic candidate or medication that slows the spread of the illness [115]. Additionally, research in the murine triple-transgenic AD models demonstrated that quercetin therapy may effectively cure pathogenic processes such as microgliosis, tauopathies, α-amyloidosis, and astrogliosis. Both memory and learning abilities in the test animals increased [113].

### 4.7. Antiviral Activity

The antiviral action of quercetin against a variety of viruses has been demonstrated. For instance, due to the lack of antiviral medications available, dengue virus infection (DENV) is still an issue in nations like Indonesia. A study was performed to evaluate the effect of quercetin on DENV. The DENV strain-2 New Guinea C-infected human cell line Huh-7 it-1 was used for the investigation. An MTT test was used, and the results showed that the quercetin IC50 and CC50 values were 18.41 g/mL and 217.113 g/mL, respectively. The findings verified quercetin’s efficacy as a DENV-2 antiviral agent [116]. The effectiveness of quercetin against human T-lymphotropic virus 1 and Japanese encephalitis virus (JEV), which is responsible for the mosquito-borne illness Japanese encephalitis, has been documented [117,118,119]. The potential anticancer effects of quercetin and isoliquiritigenin against EBV-associated gastric carcinoma (EBVaGC), as well as their antiviral effects against EBV, were investigated by M. Lee et al. [117]. The results showed that both quercetin and isoliquiritigenin were toxic to EBVaGC SNU719 cells, with quercetin demonstrating a higher induction of apoptosis and cell cycle arrest compared to isoliquiritigenin in SNU719 cells. Additionally, quercetin was found to reduce EBV latency and inhibit EBV infection, whereas isoliquiritigenin increased EBV latency. These findings suggest that quercetin shows promise as a potential lead compound for the treatment of EBV-associated gastric carcinoma. Moreover, quercetin’s ability to enhance antiviral activities with minimal side effects when combined with other supplements, such as vitamin C, has also been confirmed [120]. A derivative of quercetin, which contains a glucoside molecule, was tested for its ability to inhibit Zika virus strains in both in vivo and in vitro experiments. The results showed that Q3G exhibited cytopathic effects at concentrations of 1.2–1.3 μmol/L for EC50 and 1.5 μmol/L for EC90. The findings indicated that animals treated with Q3G showed greater survival rates and less weight loss compared to the control group [98]. It has been found that several quercetin formulations, including quercetin-3-O-D-glucuronide, quercetin-enriched lecithin formulations, and quercetin 7-rhamnoside, are effective against the influenza-A virus and porcine epidemic diarrhea virus, respectively [121]. Nevertheless, herpes simplex virus (HSV)-1, HSV-2, and drug-resistant HSV-1 were all prevented from attaching to and infecting Madin-Darby canine kidney NBL-2 (MDCK) cells by quercetin [122]. According to research conducted by Rojas et al. [95], quercetin has anti-hepatitis C (HCV) viral action. Quercetin’s antiviral effects were seen in primary hepatocytes (PHH) and Huh-7.5 cells that were infected with HCVcc at various stages of the HCV life cycle. Significant viral load reductions of up to 85% in Huh-7.5 and 92% in PHH were seen in both cell types. When quercetin was administered directly to viral particles, there was a 65% decrease in their ability to spread infection. It is interesting to note that quercetin also promoted DGAT up-regulation and the localization of the HCV core protein to the surface of lipid beads. Hence, the antiviral activities of quercetin against HCV were confirmed to be host-mediated.

### 4.8. Anti-Obesity

Quercetin, a naturally occurring flavonoid found in various plant-based foods, has been widely studied for its potential anti-obesity activity. Several in vivo and in vitro studies have shown that quercetin possesses anti-obesity properties through multiple mechanisms [123]. Quercetin has been shown to inhibit adipocyte differentiation and adipogenesis, which are critical processes in the development of obesity [124]. Moreover, it has been found to enhance lipolysis, the breakdown of stored fats, and increase thermogenesis, the production of heat by brown adipose tissue, leading to increased energy expenditure [125]. Additionally, by inhibiting the generation of pro-inflammatory cytokines, quercetin has anti-inflammatory properties, which are known to play a role in obesity-related inflammation. Ahmed et al. [126] investigated the significant roles of curcumin and quercetin in preventing obesity. It is interesting to note that both quercetin and curcumin nanoparticles outperformed their free equivalents in terms of impact, which is likely attributable to the improved solubility and bioavailability of nano-formulations. It may be possible to develop clinically practical methods to treat obese people as a result of the potential therapeutic effects of curcumin- and quercetin-modified nanoencapsulation. The antioxidant properties of quercetin are related to its beneficial effects on obesity. In fact, quercetin inhibits obesity by lowering the oxidative stress brought on by obesity and by activating the adenine monophosphate-activated protein kinase and mitogen-activated protein kinase signaling pathways, among other signaling pathways [127]. A study concluded that the water solubility and intestinal membrane permeability of quercetin were enhanced by an o/w nanoemulsion of quercetin. In comparison to quercetin dispersed in 0.3% NaCMC, an oral administration of quercetin-loaded nanoemulsion in rats increased oral bioavailability by 33.51 times while also improving pharmacokinetic characteristics. Additionally, the oral nanoemulsion system’s increased quercetin water solubility and GI absorption helped prevent weight gain and adiposity in mice treated with HFD. This led to a maximum reduction in body gain and epididymal fat mass of 23.5% and 21.2%, in comparison to the HFD-treated control group, respectively. As a result, the oral nanoemulsion system created using quercetin may be a good option for the treatment of obesity [128]. Another interesting work of research conducted by Seo et al. [129] showed quercetin’s ability to prevent the aggregation of lipids in cells, zebrafish, and mice. According to the results, quercetin inhibited the activation of factors associated to inflammation, lipogenesis, and adipogenesis in both cells and mice. The mediator of inflammation and adipogenesis, i.e., MAPK signaling, was prevented. Hence, quercetin can act as a potential therapeutic system for dealing with obesity and the inflammation brought on by obesity. P-benzoquinone, O-Phospho-L-serine, sanguisorbic acid dilactone, and Glycerophospho-N-palmitoyl ethanolamine are examples of metabolites that were affected by quercetin and associated with changes in inflammation and oxidative stress related to obesity. Authors demonstrated that the gut microbiota and metabolites are controlled by quercetin to exert its anti-obesity benefits [130]. Reviews of the relevant literature clearly demonstrate that quercetin can lower blood sugar levels and cellular insulin resistance, boost insulin sensitivity and glucose uptake, and reduce body weight [131,132]. Despite substantial research on how quercetin affects obesity, its impact on PPARγ and adipokines is unknown [129]. PPARγ has been demonstrated to be activated by quercetin in some investigations, whereas its inhibitory effects have been observed in others [133]. As a result, additional study is required to understand how quercetin regulates PPARγ in obesity. The right usage of PPARγ agonists and antagonists in those suffering from obesity with various conditions can aid scientists. These findings highlight the potential of quercetin as a promising natural compound for the treatment of obesity through its multifaceted anti-obesity activity.

### 4.9. Anti-Diabetes

Diabetes mellitus (DM), which results in protein disturbances and disturbances of carbohydrate and lipid metabolisms, is a chronic metabolic disorder. It happens due to an increase in cellular resistance to insulin or a lack of insulin secretion [134]. Obesity is usually linked with type 2 diabetes mellitus (T2DM). One of the main treatment strategies involves inhibiting the carbohydrate-hydrolyzing enzymes (alpha-amylase and alpha-glucosidase) found in the gastrointestinal tract in order to dampen the rise in blood glucose after meals [135].

The potential of quercetin to prevent different types of diabetes due to its high antioxidant and anti-inflammatory properties has been shown in several studies. Among these studies, quercetin was found to significantly inhibit both alpha amylase and alpha glucosidase activities concentration-dependently and to also prevent the oxidation of lipids in pancreatic tissue homogenates [136]. Inhibiting these two enzymes that break down carbohydrates might slow the rise in blood sugar levels, and preventing pancreatic lipid peroxidation might stop the pancreas from further suffering cellular damage that might worsen insulin production and secretion [135]. The expression of two enzymes involved in the metabolism of glucose, glucokinase and glucose-6-phosphatase, was studied by Hemmati et al. [135]. To this purpose, twenty-four adult Wistar rats were split into three groups: streptozotocin-induced diabetic rats, control, and quercetin-treated group (15 mg/kg, intraperitoneally). The results support the effect of quercetin as a dietary supplement by medical experts. A result of one study demonstrated that higher quercetin consumption was noticeably related to a lower prevalence of T2DM in the Chinese adult population. A possible benefit of quercetin in reducing the risk of T2DM was shown by this finding [10]. Diabetic osteopenia, the sixth most common consequence of diabetes mellitus, is brought on by excessive blood glucose levels that lead to oxidative stress. Abu Ayana et al. [97] employed 24 adult male rats split into three groups: diabetic-induced, control, and quercetin-treated (100 mg/kg/day) to examine the impact of quercetin on osteopenia brought on by diabetes. Diabetes was induced with streptozotocin. After 12 weeks, all of the rats were put to death, and the mandibles were processed for energy-dispersive X-ray microanalysis and scanning electron microscopy (SEM). The diabetic group’s glucose levels increased along with uneven bone surfaces, but in the group receiving quercetin treatment, both blood glucose levels and alveolar bone surfaces reverted to normal. This demonstrates quercetin’s preventive function in rebuilding bone architecture damaged by diabetes [137]. The increase in AChE activity brought on by diabetes was prevented in the cerebral cortex and hippocampus by quercetin at a level of 50 mg/kg body weight. In connection with this, quercetin reduced cholinergic signaling and aided in the recovery of lost memory in diabetic rats by restoring the function of acetylcholine (ACh) [138]. The limited solubility of quercetin in gastrointestinal fluids prevents it from being used therapeutically. Singh et al. created quercetin-loaded Soluplus micelles using the co-solvent evaporation technique to improve quercetin’s solubility profile for the evaluation of its antidiabetic benefits. Characteristics of the produced micelles were examined for in vivo and in vitro research. The entrapment effectiveness was higher than 70% despite the micelles’ size being less than 100 nm. Quercetin was released at 28.75% up to 24 h, according to in vitro findings. Blood glucose levels were noticeably lower in the group receiving quercetin-loaded Soluplus micelles, based on in vivo studies. The bioavailability profile increased to 1676% compared to a pure drug solution [101]. Nevertheless, the most prevalent metabolic problems that have a direct connection to AD in older people are obesity and diabetes. Through a number of routes, such as chronic inflammation, insulin resistance, hyperglycemia, oxidative stress, hyperinsulinemia, adipokine dysregulation, and vascular dysfunction, obesity and diabetes can result in the buildup of neurofibrillary tangles, amyloid plaques, and other signs of AD. Because of their relatively few side effects, polyphenols are now being used more often in animal models and in vitro research. Quercetin is one of the most prevalent and exhibits a variety of biological and health-improving actions in a variety of disorders. Researchers have also developed a variety of quercetin-involved nanoparticles (NPs) to address quercetin’s poor solubility and limited bioavailability [139].

### 4.10. Anti-Hypertensive

Quercetin has been extensively studied for its potential anti-hypertensive effects. Several preclinical and clinical studies have demonstrated that quercetin may have a beneficial impact on blood pressure regulation, making it a promising candidate for managing hypertension. One of the actions underlying the anti-hypertensive effect of quercetin is its ability to enhance endothelial function. Endothelial dysfunction, characterized by impaired nitric oxide (NO) production and increased oxidative stress, is a hallmark of hypertension. Quercetin has been shown to promote NO bioavailability by increasing endothelial NO synthase (eNOS) expression and inhibiting the activity of enzymes that degrade NO, such as NADPH oxidase and arginase [140]. Moreover, quercetin exerts antioxidant properties by scavenging reactive oxygen species (ROS) and decreasing oxidative stress, which can help protect endothelial cells from damage and maintain their proper function. The improvement of endothelial function by quercetin may contribute to its anti-hypertensive effects by promoting vasodilation and reducing vascular resistance, ultimately leading to lower blood pressure [141]. In addition to endothelial function, quercetin has been shown to modulate other pathways involved in blood pressure regulation. For instance, quercetin has been found to inhibit the angiotensin-converting enzyme (ACE), which plays an important role in controlling blood pressure by transforming angiotensin I to angiotensin II, a potent vasoconstrictor. By inhibiting ACE activity, quercetin can help reduce the production of angiotensin II, leading to vasodilation and lower blood pressure [142]. Quercetin has also been shown to reduce inflammation, another key contributor to hypertension, by preventing the production of enzymes and pro-inflammatory cytokines, such as cyclooxygenase-2 (COX-2) and tumor necrosis factor-alpha (TNF-α) [143]. Additionally, quercetin has been reported to have diuretic effects, increasing urine production and sodium excretion, which can help lower blood volume and subsequently reduce blood pressure. According to the study, in both primary cultures and established aortic cell lines from the inflated carotid arteries of rats, quercetin caused a selective reduction in cell viability in intimal-type VSMC as opposed to medial-type VSMC. Nuclear condensation and fragmentation patterns demonstrated that apoptosis was primarily responsible for the diminished viability. Quercetin prevented the enhanced JNK activation seen in VSMC of the intimal type. These actions might contribute to quercetin’s anti-hypertensive properties [144]. Marunaka et al. [145] claim that quercetin exerts its blood pressure-lowering effects by altering several variables that affect blood pressure, including vascular compliance (receptive to resistance and elastance); total blood volume is influenced by the autonomic nervous system, the volume of body fluids, and the renin–angiotensin system via their anti-inflammatory and antioxidant properties. The unique function of “cytosolic Cl” in quercetin’s anti-hypertensive impact on the volume-dependent control of blood pressure was also established by these scientists. Furthermore, the administration of quercetin significantly reduced blood pressure (BP) and improved endothelial dysfunction in hypertensive rats by inhibiting the endothelin-1 (ET-1)-induced enhancement of protein kinase C (PKC) activity [146]. Quercetin was found to reduce blood pressure in hypertensive rats by enhancing their antioxidant defense system. Deoxycorticosterone acetate–salt was used to generate hypertension and oxidative stress in rat models. Quercetin treatment in these rats enhanced the antioxidant defense system and reduced cardiac hypertrophy and diastolic blood pressure while restoring GSH levels in the liver and heart [147]. The impact of quercetin on hypertension in humans was investigated by Edwards et al. [146]. For 28 days, 730 mg of quercetin or a placebo was given to 19 pre-hypertensive patients and 22 subjects with stage 1 hypertension. The blood pressure of pre-hypertensive patients was not affected after quercetin treatment, whereas the blood pressure of stage 1 hypertensive subjects was. Quercetin‘s ability to reduce blood pressure was also supported by a different investigation [148]. According to a study, quercetin lowers blood pressure by a mechanism independent of the angiotensin-converting enzyme endothelin-1, or NO. Nonetheless, in the entire research group of pre-hypertension and stage 1 hypertensive overweight-to-obese participants, regular administration of a supra-nutritional dosage of 162 mg/d quercetin from onion skin extract had no effect on blood pressure and endothelial function. Moreover, quercetin was able to lower 24 h SBP in the subgroup of hypertensives, indicating a cardioprotective impact but without affecting mechanistic markers [149].

### 4.11. Antiallergic

Flavonoids hve been found to inhibit the release of histamine, a key mediator of allergic reactions, from mast cells. Mast cells are immune cells that play a crucial role in allergic responses by releasing histamine upon activation [150]. Additionally, interleukin (IL)-4 and IL-13 production (Th2-type cytokines) caused by anti-IgE antibody-stimulated receptor-expressing cells (such as mast cells or peripheral blood basophils) or allergens is suppressed by flavonoids, which also block the release of chemical mediators. Due to their inhibitory action on the activation of the aryl hydrocarbon receptor, they can also alter the development of naive glycoprotein CD4 T lymphocytes [151]. Quercetin has been shown to decrease mast cell degranulation and subsequent histamine release, thereby attenuating the allergic response. This has been demonstrated in in vitro and in vivo studies, indicating that quercetin can potentially alleviate the symptoms of allergies such as itching, sneezing, and nasal congestion. Moreover, it has been claimed that quercetin possesses anti-inflammatory properties, which can be beneficial in allergic conditions. Allergic reactions are often characterized by inflammation in affected tissues. Nevertheless, quercetin has been utilized to inhibit the production of inflammatory mediators, such as prostaglandins and leukotrienes, which are involved in the inflammatory response. This anti-inflammatory property of quercetin can help reduce the severity of allergic reactions and alleviate symptoms associated with allergies.

In lung tissue, over 25% of the quercetin absorbed is localized, according to rat tests conducted on animals, which is an extra advantage in the fight against allergies and the related condition asthma [152]. Through the use of rat mast cells (RBL-2H3 cells), Sato et al. [151] evaluated the anti-allergic activity (type I hypersensitivity) of several onion cultivars. It was discovered that the cultivars’ anti-allergic potency varied greatly [153]. Indicated by the greatest inhibitory concentration value (IC50 = 89.1 mg/mL) among the investigated types, the Satsuki cultivar showed the greatest activity. Components of onion extracts like quercetin 4′-glucoside were shown to be positively correlated (r = 0.91) with an anti-allergic reaction [154]. In a mouse model of asthma, T. T. Oliveira et al. [155] investigated the effects of quercetin and onion extract on cytokines and smooth muscle contraction in vitro. They observed a decrease in a release of tracheal rings, production of inflammatory cytokines, a reduction in the number of cells in bronchoalveolar lavage, and a drop in eosinophil peroxidase in the lungs after treatment with quercetin or onion extract.

Disodium cromoglycate, an anti-allergic medication, and quercetin exhibit similar structures. On cultured human mast cells, Weng et al. [156] evaluated the effects of quercetin and cromolyn. Both substances may successfully suppress the release of leukotrienes and histamine from primary human cord blood-derived cultured mast cells that have been activated by IgE/anti-IgE at a concentration of 100 M. Quercetin decreases IL-6 in a dose-dependent way, slows the rise in cytosolic calcium levels, and inhibits IL-8 more effectively than cromolyn does.

Food allergies comprise a serious and prevalent health issue that is on the rise. The only effective allergy treatment is the avoidance of trigger foods [157]. Since there is such a heavy load on young children, studies of this issue focus particularly on prevention [158]. Inhibiting the action of dendric cells is a key approach in the treatment of food allergies. To control mucosal immunity during an allergic reaction, quercetin acts in conjunction with kaempferol and isoflavones. In a rat model using rats sensitized by peanut infusion, Shishehbor et al. [159] examined quercetin’s effect on peanut-induced anaphylaxis responses. Peanut-induced anaphylaxis responses were completely stopped after four weeks of regular quercetin consumption. The values of plasma histamine in rats receiving quercetin were significantly lower compared to those of rats in the positive control group, according to the study. Quercetin can, therefore, be used as an alternative treatment to protect against IgE-mediated food allergies since it is a powerful inhibitor of ongoing immunoglobulin E reactions against peanut proteins.

### 4.12. Anti-Asthmatic

The complicated inflammatory illness known as asthma is characterized by eosinophilic inflammation, excessive goblet cell mucus output, and airway inflammation and hyperresponsiveness [160]. Wheezing, airway hyperresponsiveness, and airway obstruction are common symptoms of asthma. It is distinguished by airway edema, increased mucus secretion brought on by the airway, smooth muscle hypertrophy/hyperplasia, recruitment of eosinophils, systemic immunoglobulin E (IgE) production, airway hyperresponsiveness to allergens, and certain leukocytes [161]. Quercetin, a natural flavonoid, has been shown to have anti-asthmatic effects. Quercetin possesses potent anti-inflammatory and antioxidant properties that have been found to suppress the production of inflammatory cytokines that play a crucial role in asthma pathogenesis [162]. Studies have shown that quercetin can reduce airway hyperresponsiveness, eosinophilic inflammation, and mucus hypersecretion, which are the major characteristics of asthma [163]. Human airway epithelial (HBE16) cells were used to examine the influence of quercetin on MUC5AC expression caused by human neutrophil elastase (HNE) and its molecular processes. Quercetin and HNE were used to pretreat HBE16 cells. Findings indicate that quercetin can prevent the protein kinase C (PKC), epidermal growth factor receptor (EGFR), and extracellular-regulated kinase (ERK) signal transduction pathway responsible for HNE-induced MUC5AC expression in human airway epithelial cells. Quercetin’s significant inhibitory dosage was 40 M [164]. The effectiveness of *Elaeagnus pungens* leaves in the treatment of asthma and chronic bronchitis was confirmed. The major effective ingredient in *Elaeagnus pungens* leaves was quercetin. They claimed that quercetin and kaempferol have applications in asthma and chronic bronchitis treatment [165]. Sozmen et al. [166] examined the effect of quercetin (16 mg/kg/day) on histological features as well as airway epithelium in allergic airway inflammation in BALB/c mice. Quercetin treatment resulted in a reduction in epithelial thickness, subepithelial smooth muscle thickness, goblet cell count, and mast cell count when compared to untreated mice with allergic airway inflammation. According to these results, quercetin ameliorates chronic histopathological alterations in lung tissue, with the exception of basal membrane thickness, and its anti-inflammatory actions may be attributed to epithelium-derived cytokine modulators and epithelial apoptosis [166]. The effect of quercetin glycosides in neonatal asthmatic rats was investigated. Quercetin significantly decreased iNOS protein expression. Neonatal asthmatic rats showed constriction of blood vessels and airways as well as an accumulation of eosinophils in the lungs. However, quercetin glycoside treatment significantly decreased eosinophil inflammation and infiltration. In conclusion, the protective benefits of quercetin glycosides against rat neonatal asthma were demonstrated by dramatically reduced levels of inflammatory markers [167]. Luo et al. [168] investigated the effects of several herbs. According to the study, quercetin may be utilized to create novel bronchodilators that may be used to treat obstructive lung conditions including asthma and chronic obstructive pulmonary disease. The diverse effects of quercetin, including its inhibitory effect on mast cell activation, eosinophil activation, relaxation of the tracheal ring, decrease in IL-4 and IgE serum levels, and blocking of IL-8 and MCP-1 expression in airway epithelial cells, point to a valuable role of quercetin in allergic asthma. Additionally, it has been proposed that quercetin might be a therapeutic option for treating allergic asthma and shines new light on the immunopharmacological function of quercetin [163].

## 5. Bioavailability of Quercetin

The quantity of a chemical that reaches the intended place of action is known as bioavailability. Nevertheless, in the case of polyphenol absorption, bioavailability is considered as the quantity present in plasma. In addition, it is an assessment of the dose present according to urine measurement of its components and metabolites. Its equivalent phrase, ADME (absorption, metabolism, disposal, and excretion), is used in the pharmaceutical sector as a comparable term [169]. Quercetin dose and bioavailability from diverse agricultural products have been examined in recent years [170,171]. However, its bioavailability, or the amount of ingested compound that reaches systemic circulation, is limited due to several factors (Figure 5). First, quercetin is poorly soluble in water, which affects its absorption in the small intestine. Second, its extensive metabolism in the liver and intestinal epithelium decreases its bioavailability by transforming it into metabolites with lower biological activity. Third, the intestinal microbiota can also influence the absorption and metabolism of quercetin, further complicating its bioavailability [172].

Animal research can be beneficial, but due to variations in endogenous metabolism and gut bacteria, they may provide results that are challenging to transfer to human studies [173]. The gold standard for research is on humans, and these investigations often involve volunteers who have been on a low-(poly)phenol diet for around two days, taking an acute dose of food or beverage. Investigations have included ileostomy patients who have had their colon surgically removed as well as those with an intact colon; the investigates have subsequently been substantiated by analyses of ileal effluent, urine, and plasma [174]. For short-term ingestion, quercetin-rich sources had overall plasma concentrations of 72 to 193 nmol/L of both free and conjugated quercetin, whereas the human body is capable of ingesting around 20 mg of quercetin per day. Nevertheless, long-term dietary consumption does not lead to plasma accumulation [175]. Due to poor solubility, oral bioavailability, which is estimated to be 1% in humans, is also regarded to have very low absorption in the gastrointestinal system [176].

Several studies have compared quercetin absorption from different sources to determine which sources are the best for optimal absorption. In the first study, the absorption of quercetin from tea and onions was compared in healthy participants. Participants ingested 129 g of fried onions per day and 1600 mL of black tea per day (each containing 49 mg of quercetin glycosides), for a total of three days, followed by a four-day washout period. The results of the study showed that urinary excretion of quercetin was higher after consumption of fried onions compared to black tea, suggesting that the form of quercetin present in onions is better absorbed than that in tea [177]. Similarly, in another study [178], an investigation was conducted to quantify quercetin absorption from the small intestine only, without considering microbiome metabolism, using onions, rutin, and aglycones as sources of quercetin. The results showed that quercetin glucosides were found to have the highest intestinal absorption in comparison with quercetin rutinosides and aglycones. A recent study [179] investigated whether the placement of the glucose moiety impacted absorption. The results suggested that there was no difference in absorption between quercetin-3-glucoside and quercetin-4′-glucoside, suggesting that the glucose attachment on positions 3′ and 4′ had no effect on the absorption rate. The proposed transport mechanism in the small intestine may partially be responsible for this [174]. Finally, a study by Guo et al. [180] studied the impact of dietary fat on quercetin absorption. A comparison of a fat-free trial with a low-fat trial and high-fat trial resulted in increases in total plasma quercetin concentration of 12% and 17%, respectively. This suggests that dietary fat can increase the absorption of quercetin [180]. Overall, these studies demonstrate that the form of quercetin and dietary factors such as fat intake can impact the absorption of this important flavonoid. Toward absorption, quercetin is transferred to the liver where it passes through phases I and II of metabolism to produce metabolites that are distributed to body tissues via circulation. Understanding the main metabolites in the urine and blood is crucial for recognizing quercetin bioavailability [170].

### Improvement of Quercetin Bioavailability

As previously mentioned, quercetin’s weak water solubility and chemical stability frequently place restrictions on its bioavailability. Therefore, to enhance the bioavailability of quercetin, several types of encapsulation techniques can be used (Figure 4). Encapsulation is a process in which droplets are covered by a coating and embedded in a homogeneous or heterogeneous matrix to create small capsules that act as a physical barrier between the product’s other components and the core [181,182,183]. In order to enhance a substance’s functioning, handling, dispersibility, stability, and/or performance, encapsulation involves trapping an active agent often enclosed within another substance [184,185,186].

Quercetin has been delivered using liposomes. Phospholipids, which have two non-polar hydrophobic hydrocarbon tails and a hydrophilic polar headgroup, are frequently used for creating liposomes. Due to their hydrophobic properties, phospholipids tend to connect with one another when distributed in water, resulting in the creation of bilayer structures. These bilayers are capable of fusing with one another to create spherical shells with a watery core encircled by one or more bilayers. Hydrophobic compounds (like quercetin) can be enclosed inside the non-polar sections of the phospholipid bilayers, whereas hydrophilic molecules (like water) can be encased within the watery core [187]. The potential of liposomes to enhance the bioactivity and bioavailability of quercetin has been the subject of several investigations [188,189,190]. It has been demonstrated that liposomes offer defense against the stomach’s enzymes and acids [191]. Nevertheless, since bile salts are present and phospholipases hydrolyze phospholipids, they might be broken down in the small intestine [192]. Additionally, quercetin-loaded liposomes have been trapped within chitosan microgels, which improved the bioavailability of the compound in comparison to plain liposomes alone under simulated gastrointestinal tract conditions [193].

For the delivery of quercetin, emulsions have also been employed. Emulsions are a type of colloidal thermodynamic system that is unstable and shows destabilization characteristics [194]. There are several emulsion types that may be employed to encapsulate quercetin, but oil-in-water (O/W) emulsions are the most widely utilized. For some applications, nanoemulsions are preferable over emulsions for a number of reasons: By lowering the mean droplet diameter below 50 nm, optically transparent systems can be produced. A major benefit to encapsulating hydrophobic substances in this way is their strong resistance to gravitational separation (creaming or sedimentation), aggregation (flocculation or coalescence), and digestion in the gastrointestinal tract, which increases their bioavailability [195]. The rice bran protein has been used to stabilize an oil-in-water emulsion in order to encapsulate quercetin [196]. Researchers demonstrated that amounts of quercetin up to 5 mg/mL could be effectively encapsulated at a high encapsulation efficiency (98%) into nanoemulsions. In addition, the authors noted that after simulating digestion, the bioavailability of quercetin in its encapsulated form (12.7%) was approximately ten times more than that of the non-encapsulated form (1.4%). After digestion, mixed micelles were thought to have formed in the gastrointestinal fluids, solubilizing the quercetin and preventing its degradation [196]. Curcumin and resveratrol, two additional polyphenols, were compared to quercetin’s in vitro gastrointestinal stability. The polyphenols’ oil–water partition coefficients have been demonstrated to be correlated with their stability; the more hydrophilic the molecule, the longer it remains in the gastrointestinal fluids and the greater the amount of chemical degradation or precipitation. Due to the lower oil–water partition coefficient (logD) of quercetin (2.17) compared to resveratrol (3.39) or curcumin (4.12), there was a smaller fraction of quercetin (44%) remaining after simulating digestion than curcumin (92%) or resveratrol (100%). On the other hand, when the lipid phase was digested, the bioaccessibility of the quercetin was rather high (>70%) [197]. A different study by Zhou, Zheng, and McClements (2021) revealed that the kind of oil (triglyceride oils made up of either long-chain or medium-chain fatty acids) affected the bioaccessibility of quercetin and gastrointestinal stability, emphasizing the significance of picking a suitable oil phase [198]. The efficiency of quercetin delivery via emulsions and nanoemulsions has also been investigated as part of in vivo studies [199]. For instance, as a treatment for diabetes, oil-in-water nanoemulsions containing quercetin have been stabilized with a food-grade non-ionic surfactant called Tween 80. As a result of orally administering these quercetin-loaded nanoemulsions to rats, the authors investigated blood glucose levels, oxidative stress, and inflammatory markers. According to their findings, the quercetin-loaded nanoemulsions were able to lower these indicators, highlighting a possible therapeutic benefit.

Another technique for enhancing the solubility, stability, and bioavailability of bioactive chemicals is cyclodextrin complexation. The cyclic oligosaccharides that make up cyclodextrins (CDs) are produced when starch is treated enzymatically [200,201]. There are several types of CD that may be used in culinary applications, including α-, β-, and γ-forms. These CDs differ in the quantity of glucose units they contain, which affects the molecular size of their hydrophobic cavity [202]. Hence, the cavity size must be chosen in accordance with the dimensions of the non-polar molecule in order to encapsulate it [203]. In order to improve the cavity size of β-CDs, Park et al. [204] produced mono-6-deoxy-6 aminoethyl amino-cyclodextrin (Et- β-CD), which increased the quantity of quercetin that could be dissolved in water by around 35-fold while enhancing its antioxidant activity and photostability. Furthermore, a 22-fold increase in solubilization efficiency was noted when compared to the β-CD complex. The water-solubility of quercetin has been improved by encapsulating it inside of β-CDs utilizing a solvent evaporation method. However, it was noted that the quantity of quercetin that could be encapsulated was constrained by the relatively small cavity size of β-CDs [205]. The casein nanoparticles utilized to encapsulate the quercetin–HP–β-CD complex postponed its release under gastrointestinal circumstances. As a consequence, until it reached the gut epithelium, the quercetin was able to move through the digestive tract with low loss. Rats were given the formulations orally, and it was discovered that the encapsulated quercetin had a ninefold greater total bioavailability (37%) compared to the control samples [206]. According to a similar study, quercetin content in the blood plasma (Cmax) increased after oral treatment (up to 176 g/mL) when a quercetin–HP–CD complex was encapsulated into zein nanoparticles [207].

The stability and bioavailability of quercetin can also be improved by encapsulating it within hydrogels. Typically, polysaccharides that have been crosslinked chemically or physically are used to assemble these hydrogels [208]. A porous 3D polymer network makes up the interior of hydrogels, which traps an enormous quantity of water (typically more than 90%) [209]. Both of the following major mechanisms—(a) strong attractive interactions with the polymer network; and (b) encapsulating hydrophobic bioactive substances among the hydrophobic areas of colloidal particles before incorporating them into the polymer network—can trap hydrophobic bioactive substances in these interiors [210]. The quercetin-loaded colloidal particles in the latter scenario must be greater than the pore size of the polymer network to prevent fast release. Microgels, which are made up of tiny hydrogel beads that are spherical in shape, are often utilized for encapsulation and distribution [211]. The encapsulation of quercetin in gellan gum microgels demonstrated its protection from deterioration under simulated gastrointestinal conditions [212]. Moreover, an in vivo animal study has been performed using mice to study the bioavailability of quercetin encapsulated in whey protein microgels coated with lotus root amylopectin (LRA) [213]. It was noted that mice given the quercetin-loaded microgels had serum levels of quercetin approximately eight times higher than mice fed quercetin alone. Microgels made of pectin and oligochitosan have also been used to encapsulate quercetin [214]. Due to the presence of pectinases, it was demonstrated that these microgels remained stable under simulated gastric and intestinal fluids but broke down under simulated colonic fluids. By adjusting the pectin content in the microgels, the amount of quercetin produced in the simulated colonic fluids could be managed. In order to preserve quercetin in the upper gastrointestinal tract while releasing it in the colon, this type of microgel technology is appropriate.

## 6. Conclusions

Quercetin, a remarkable flavonoid with potent antioxidant and anti-inflammatory properties, is expected to revolutionize the pharmaceutical industry as a multi-faceted therapeutic agent. The extensive research on quercetin has unveiled a multitude of health benefits, offering a promising outlook for various medical conditions. As advancements in science and technology continue, the precise mechanisms underlying quercetin’s effects are becoming clearer. Its ability to scavenge free radicals and combat oxidative stress makes it a potential shield against age-related diseases and other conditions influenced by cellular damage. Furthermore, its proven anti-inflammatory properties, demonstrated through the inhibition of inflammatory cytokines and enzymes, pave the way for its application in treating inflammatory conditions like asthma, arthritis, and inflammatory bowel disease. Medical professionals and researchers are eagerly awaiting the outcomes of ongoing clinical trials, hopeful that quercetin could become a mainstay in inflammation management. But the potential of quercetin goes beyond its antioxidant and anti-inflammatory effects. Emerging evidence suggests that quercetin may be a potent player in the fight against viral infections. Its antiviral properties, which extend to combatting influenza, hepatitis B and C viruses, and HIV, make it an appealing candidate for future antiviral therapies. Researchers are hopeful that with further development, quercetin-based medications could revolutionize the treatment of viral diseases, potentially reducing the burden of global health crises. One of the most exciting revelations is quercetin’s anticancer potential. Studies have shown that it can hinder cancer cell proliferation and trigger apoptosis in various cancer types, including breast, lung, prostate, and colon cancers. As we look forward to the future, quercetin-based cancer therapies may form part of a comprehensive approach to target and combat cancer, providing new hope to patients battling this formidable disease. The cardiovascular benefits of quercetin are also an area of great promise. Its ability to lower blood pressure, reduce cholesterol levels, and improve endothelial function showcases its potential in the prevention and treatment of cardiovascular diseases. As we seek solutions for heart-related ailments, quercetin may become an essential component of future therapies aimed at improving heart health and reducing the prevalence of cardiovascular disorders. To ensure the widespread and effective use of quercetin as a pharmaceutical agent, researchers are diligently working on enhancing its water solubility, chemical stability, and bioavailability. Various delivery systems, such as lipid-based carriers, polymer nanoparticles, inclusion complexes, micelles, and conjugate-based systems, are being explored to optimize quercetin’s therapeutic potential. These cutting-edge techniques hold great promise for future drug development and personalized medicine, as they can be tailored to suit specific routes of administration, target tissues, and safety considerations. In the coming years, the integration of quercetin-rich foods into daily diets and the development of specialized quercetin supplements will likely become common practices to maintain overall health and prevent chronic diseases. This proactive approach to healthcare could lead to a paradigm shift, focusing on holistic well-being and preventative measures rather than solely treating symptoms after they arise. As researchers continue to explore the potential applications of quercetin, collaboration between pharmaceutical companies, academic institutions, and regulatory agencies will play a vital role in turning these promising findings into tangible medical breakthroughs. The future of quercetin in pharmaceuticals is undoubtedly bright, and its versatile properties as an antioxidant, anti-inflammatory, antiviral, anticancer, and cardiovascular agent make it a beacon of hope in the quest for improved healthcare and disease management.

## Figures and Tables

**Figure 1 pharmaceuticals-16-01020-f001:**
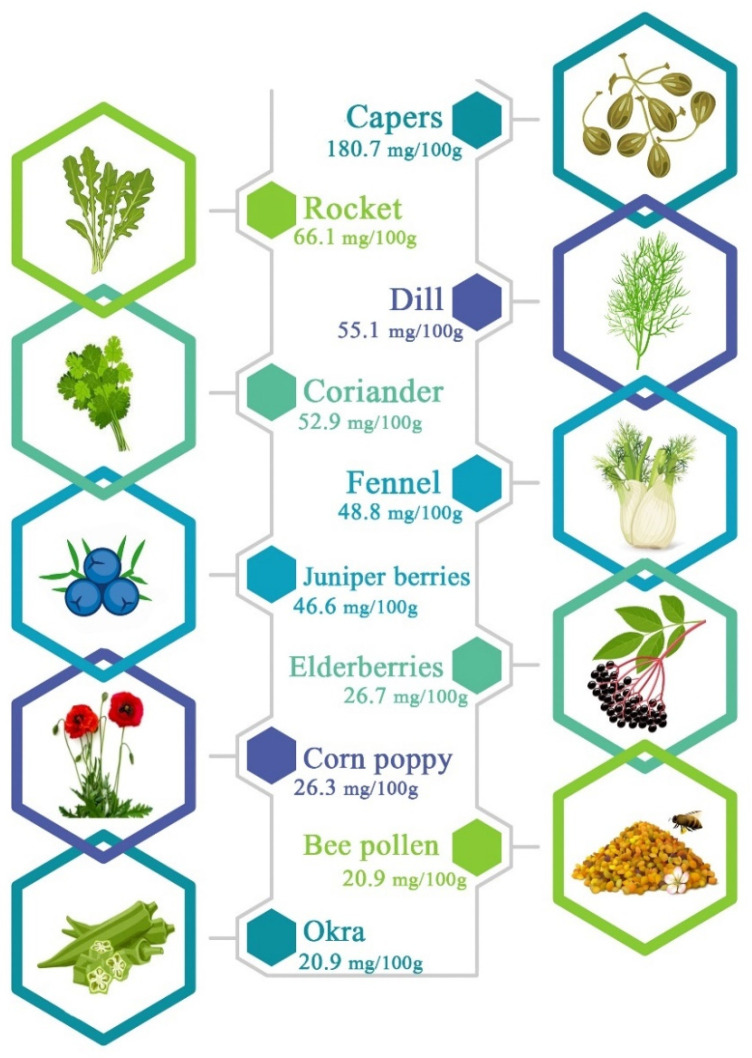
The main sources of quercetin [4].

**Figure 2 pharmaceuticals-16-01020-f002:**
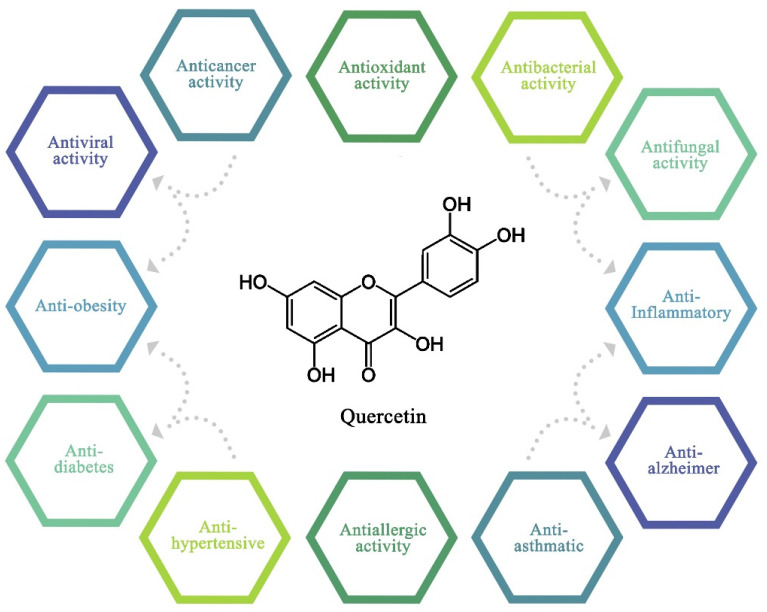
Structure and potential applications of quercetin.

**Figure 3 pharmaceuticals-16-01020-f003:**
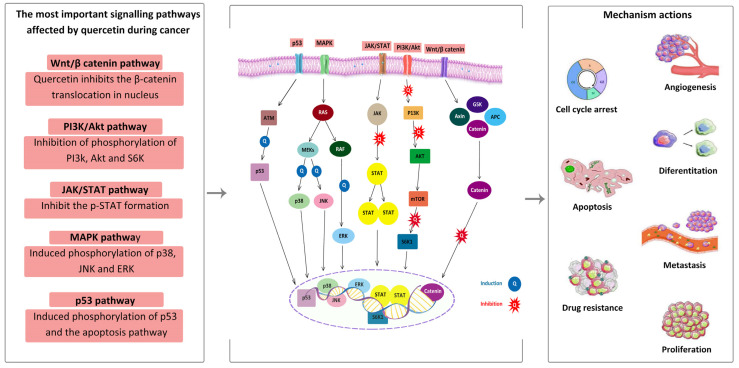
Quercetin activity in different cancer types.

**Figure 4 pharmaceuticals-16-01020-f004:**
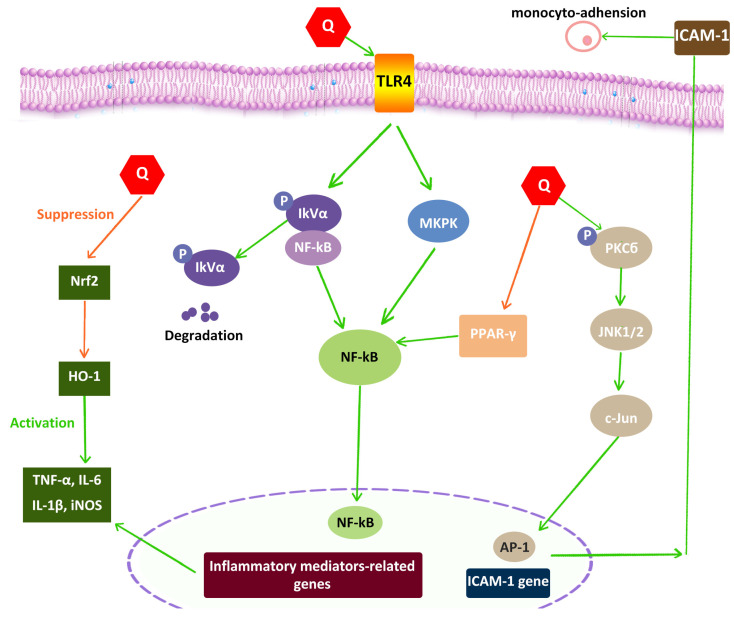
The anti-inflammation signal pathways of quercetin. Abbreviations: PPAR-γ, peroxisome proliferator-activated receptor-γ; MAPK, mitogen-activated protein kinase; JNK, c-Jun N-terminal kinases; ICAM-1, intercellular adhesion molecule-1; Nrf2, nuclear factor erythroid 2-related factor 2; PKCδ, protein kinase c-delta; TNF-α, tumor necrosis factor-α; iNOS, inducible nitric oxide synthase; IL-6, interleukin-6; IL-1β, interleukin-1β; AP-1, activator protein 1; HO-1, heme oxygenase-1; NF-κB, nuclear factor-kappa B; IκBα, nuclear factor-kappa B inhibitor.

**Figure 5 pharmaceuticals-16-01020-f005:**
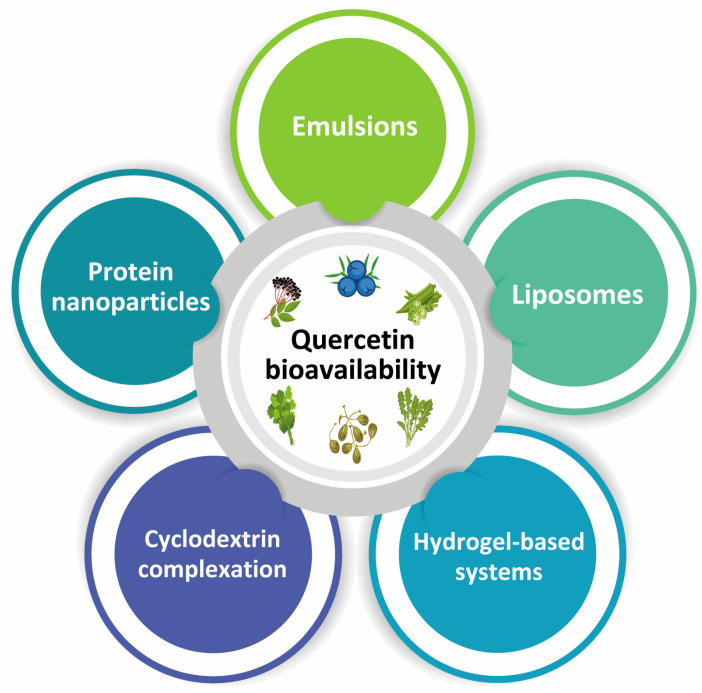
Recent strategies on the enhancement of quercetin bioavailability.

**Table 1 pharmaceuticals-16-01020-t001:** Antioxidant properties of quercetin.

Antioxidant Agent	Activity Carried Out	Key Findings	Study Models	References
Quercetin	Aging ovary-oxidative stress	Related genes are more prevalent	Animal models/in vivo	[30]
Quercetin	H_2_O_2_	Quercetin has neuroprotective properties in addition to reducing the cytotoxicity of H_2_O_2_	Clinical trial/in vitro	[23]
Onion extract plus six different types of quercetin	DPPH	There was an increase in the antioxidant activity of quercetin-3-O-glucuronide, isorhamnetin, tamarixetin, and quercetin	In vitro	[19]
Quercetin with Hesperidin	Hydrogen peroxide radical, DPPH, hydroxyl radical, nitric oxide, reducing power assay, and superoxide	Enhanced effectiveness of both substances against DPPH	In vitro	[18]
Quercetin and its glucosides	Hydroxyl groups’ function in DFT	The antioxidative exercises are predominantly OH, groups in the C-ring and B-ring	In vitro	[31]
Quercetin	Oxidative stress and BP in factors in urine and blood	No effects on oxidative stress; BP reduction	Clinical trial/ex vivo	[32]
Quercetin	Anti-reactive oxygen species (ROS)	GSH level regulated; decreases MDA levels while increasing SOD activity	Animal model/in vivo	[33]
Quercetin	Anti-free radical	Scavenging DNA from the free radicals	In vitro	[34]
Quercetin	Anti-aging	Reduced the severity of mouse senescence and lengthened the life of *C. elegans* by 15%	Animal model/in vivo	[35]
Quercetin with Na^+^, K^+^-ATPase	Anti-aging in humans	Influences the activity of ATP-dependent protein transporter	Clinical trial/in vitro	[36]
Quercetin	Cancer treatment	The antioxidant activity of quercetin caused inhibition of tumor growth	Milad	[37]
Quercetin	Anti-aging	Prevention of developing eye diseases such as macular degeneration and cataracts	Clinical trial/in vitro	[38]
Quercetin	Antidepressant activity	Increased 5-HT levels and prevented MAO-A levels in brain	Animal model	[39]
Quercetin	Antidepressant activity	Quercetin treatment with a concentration of 50 mg/kg for 8 weeks restored the levels of serum elements	Animal model	[40]
Quercetin	Antidepressant activity	Lipid hydroperoxide levels in the hippocampus rose after olfactory bulbectomy, proving improved depression in rats	Animal model	[41]
Quercetin	Anti-non-alcoholic liver	Liver protection from NASH by decreasing the levels of CYP2E1	Animal model	[42]
Quercetin	Anti-free radical	Maintains homeostasis in the human body by removing free radicals	In vivo	[43]
Quercetin	Fatty liver treatment in non-alcoholics	Prevention of NAFLD and liver steatosis	In vivo	[44]
Dihydroquercetin	Renal protection	Regulating the nuclear factor 2 (Nrf2) signaling pathways connected to erythroid 2 (Nrf2).	Animal model/in vivo and in vitro	[45]
Quercetin	Quercetin treatment in kidney and tumor tissues	Regulated mitogen-activated protein kinases (MAP kinases)	Animal model/in vivo	[46]
Quercetin	Kidney disorder	Negatively regulated the phosphatidylinositol 3-kinase (PI3K/Akt).	Animal model/in vitro	[47]
Quercetin	kidney fibrosis treatment	Activated B-cell nuclear factor kappa light-chain enhancer hedgehog	Animal model/in vivo and in vitro	[48]
Quercetin	Anti-inflammatory	Participated in the overall biological activity of a quercetin-rich diet	Ex vivo	[20]

**Table 2 pharmaceuticals-16-01020-t002:** Anticancer properties of quercetin.

Cancer Type	Mechanisms	References
Lung cancer	BCL2/BAX-mediated necrosis, apoptosis, and mitotic catastrophe are all triggered Inhibits the ability of A549 cells to migrate;	[57]
Lung cancer	increases miR-21 expression and inhibits [Cr(VI)]	[58]
Lung cancer	In nanoparticle form, improved antitumor effect	[59]
Gastric cancer cell lines	Combination treatment results in improved anticancer effects	[60]
13 HCC liver cancer cell line	Both substances’ combined anticancer effects	[61]
Ovarian cancer mouse model	Hydrogel of quercetin demonstrates improved apoptosis	[62]
EMT6 breast cancer cell line	In combined form, synergistic antitumor effects	[63]
Breast cancer	Slows cell cycle progression and increases cell apoptosis; p51, p21, and GADD45 signaling activity are upregulated, as is FasL mRNA expression	[64]
Colon cancer	ERK activation triggers G2 phase arrest and autophagic cell death	[65,66]
Kidney cancer	Increases the cytotoxic effects of DOX and guards against nephrotoxicity caused by DOX; reduces the expression of TNF, IL1B, iNOS, and caspase-3 in the kidneys	[67]
Thyroid cancer	Reduces chymotrypsin-like proteasome activity; raises the rate of apoptosis and decreases the rate of cell proliferation via activating caspases; reduces the concentration of Hsp90	[68,69]
Eye cancer	Reduces the secretion of VEGF, RPE cell proliferation, and migration dose-dependently; VEGF production generated by CoCl2’s hypoxia is prevented	[70]
Brain cancer	Hsp27 inhibition reduces COX2 expression and serves as both a COX2 and Hsp27 inhibitor	[71]
Neck and head cancer	HSC3 cell colony growth is reduced; retards MMP2 and MMP9 levels	[72]
Gastric cancer	EBNA1 and LMP2 protein expression from EBV is inhibited	[55]
Prostate cancer	Prevents vimentin and N-cadherin expression that is stimulated by TGF-β; reduces Twist, Snail, and Slug expression when TGF-β is present in the prostate cancer 3 cell line	[73]
Pancreatic cancer	Cellular FLICE-like inhibitory protein expression is decreased	[74]
Skin cancer	Blocks the NF-kB activation and COX-2 up-regulation caused by UVB light in the Hacat cell line	[75]
Bone cancer	Reduces the expression of cyclin D1 in U2OSPt and SKOV3 cells	[76]
Liver cancer	Induces apoptosis	[52]
Breast cancer	Increases the amounts of cleaved caspases 8 and 3; inhibits the manufacture of phosphorylated JAK1 and STAT3; -reduces the activity of the STAT3-dependent luciferase reporter gene in BT474 cells.	[77]

## Data Availability

The data presented in this study are available in this article.

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
