# Peer review of "Recent Advances in Potential Health Benefits of Quercetin"

_pharmaceuticals, 2023, doi:10.3390/ph16071020_

Round 1

Reviewer 1 Report

The article is very interesting before, however, before it could be considered for publication authors need to incorporate certain changes and revise the manuscript accordingly.

Abstract section does not give proper information. The abstract has been written with respect to its anti-inflammatory, anticancer, and cardioprotective effects, but at the end, authors mentioned different parameters covered in the paper. It is better to refine the abstract. Also, highlight essentialities and future perspectives of the study.

In the introductory section,

The introductory section is written in a good way with two figures. It would be better if authors remove the words quercetin sources from the figure.

In the chemical structure and main sources section

Authors should note that plant species name should be written in italics. e.g., in the section, “More than twenty plant species, including Santalum album, Mangifera indica, Emblica ofcinalis, Curcuma domestica valenton, Withania somnifera, Foeniculum vulgare, and Cuscuta refexa contain quercetin”

In the Antioxidant activity section

Authors should take in to consideration of reference writing style for the journal e.g., Lesjak et al. (2018)

In Table 1,

Authors should clearly mention what “Medical applications” mean? Additionally, C. elegans should be written in italics.

In anticancer activity section

Some sentences are written without any reference to literature e.g., According to the study, administering nano-quercetin to MCF 7 breast cancer cells increases mRNA expression and apoptosis. Similar problem exists at subsequent sections also.

In Table 2,

Authors have written “-Increases miR-21 expression and inhibits [Cr(VI)]”. I fail to understand relevance of chromium and lung cancer as part of the mechanism.

In antibacterial and antifungal section, references style of the journal needs to be followed.

In Table 3,

A. niger needs to be in italics. The table contains serious flaws under “Type of cancer” and “activity performed” headings. Antibacterial is not any type of cancer and bacterial name is not any type of activity.

In Bioavailability of Quercetin section, it is advised to add some suitable figure.

Discussion

Discussion is needed to be taken care of? Please update your discussion in the light of recent reports from 2019-23.

Needs improvement

Author Response

The article is very interesting before, however, before it could be considered for publication authors need to incorporate certain changes and revise the manuscript accordingly.

We would like to thank you for the time and effort taken to review our manuscript, we have carefully responded to each comment. Your requested corrections carried out in this second draft are in track changes.

Abstract section does not give proper information. The abstract has been written with respect to its anti-inflammatory, anticancer, and cardioprotective effects, but at the end, authors mentioned different parameters covered in the paper. It is better to refine the abstract. Also, highlight essentialities and future perspectives of the study.

The abstract has been refined and the future prospective has been included.

In the introductory section,

The introductory section is written in a good way with two figures. It would be better if authors remove the words quercetin sources from the figure.

The words “quercetin sources” have been deleted from the figure.

In the chemical structure and main sources section

Authors should note that plant species name should be written in italics. e.g., in the section, “More than twenty plant species, including Santalum album, Mangifera indica, Emblica ofcinalis, Curcuma domestica valenton, Withania somnifera, Foeniculum vulgare, and Cuscuta refexa contain quercetin”

All the scientific names were changed to italics.

In the Antioxidant activity section

Authors should take in to consideration of reference writing style for the journal e.g., Lesjak et al. (2018)

There were a few citations that needed to be changed, and they all changed to the reference style of the journal.

In Table 1,

Authors should clearly mention what “Medical applications” mean? Additionally, C. elegans should be written in italics.

All "Medical applications" changed to reflect the exact aim of the research, and the word has been changed to italic.

In anticancer activity section

Some sentences are written without any reference to literature e.g., According to the study, administering nano-quercetin to MCF 7 breast cancer cells increases mRNA expression and apoptosis. Similar problem exists at subsequent sections also.

The citations have been added to the manuscript.

In Table 2,

Authors have written “-Increases miR-21 expression and inhibits [Cr(VI)]”. I fail to understand relevance of chromium and lung cancer as part of the mechanism.

There is no direct evidence to suggest that chromium itself causes lung cancer. However, specific forms of chromium compounds have been associated with an increased risk of lung cancer, especially in certain occupational settings where exposure levels may be higher. There are two primary forms of chromium that are relevant to this discussion:

  1. Hexavalent chromium (Cr(VI)) (That we mentioned): This is the more toxic form of chromium and is known to be a human carcinogen. Occupational exposure to hexavalent chromium occurs in industries like stainless steel welding, chromate production, and other processes involving chromium metal and compounds. Inhalation of hexavalent chromium can lead to an increased risk of lung cancer, particularly among workers with prolonged and significant exposure.
  2. Trivalent chromium (Cr(III)): Trivalent chromium is the naturally occurring form of the element and is generally considered less toxic than hexavalent chromium. It is an essential trace element involved in certain metabolic processes. Trivalent chromium is not known to be a direct cause of lung cancer.

In antibacterial and antifungal section, references style of the journal needs to be followed.

In all parts of the article, the citations followed the journal's reference style.

In Table 3,

  1. niger needs to be in italics. The table contains serious flaws under “Type of cancer” and “activity performed” headings. Antibacterial is not any type of cancer and bacterial name is not any type of activity.

The word has been changed to italic, and the name of the column has been changed to “biological activity”.

In Bioavailability of Quercetin section, it is advised to add some suitable figure.

The figure has been added to this section.

Discussion

Discussion is needed to be taken care of? Please update your discussion in the light of recent reports from 2019-23.

The conclusion has been changed and updated to pay more attention to quercetin's characteristics and health benefits.

Reviewer 2 Report

The authors decided to write a narrative review on the health benefits of quercetin according to the title. This manuscript does not address a new specific gap in the field since similar titles have been updated recently. There are fundamental problems in the bibliography, as mentioned in the comments. The recently published materials on this title are ignored by the authors, which is totally against publication ethics. Besides declaring the requirement of a new submission (despite the existence of similar titles from 2020-2022) the authors need to work hard on the literature survey, appropriate figures, and integrated discussion to organize the manuscript. Otherwise, the manuscript cannot be further considered in its current status.

Comment 1) The authors should define the new aspects of their manuscript compared to similar articles published recently and the reason for their ignorance. These are only some examples:

●Septembre-Malaterre et al. Focus on the high therapeutic potentials of quercetin and its derivatives. Phytomedicine Plus. 2022 Jan 15:100220.

●Deepika, Maurya PK. Health benefits of quercetin in age-related diseases. Molecules. 2022 Apr

●Salehi et al. Therapeutic potential of quercetin: New insights and perspectives for human health. Acs Omega. 2020 May 14;5(20):11849-72.

Comment 2) There are lots of published papers on quercetin therapeutic potentials with appropriate inclusion criteria, which have been neglected without a known reason. What were the keywords, databases, range of years etc… to consider an article? The authors should limit their investigation to the recent three years and update their manuscript for neglected data. These are only some examples: 

●Nutmakul T. A review on benefits of quercetin in hyperuricemia and gouty arthritis. Saudi Pharmaceutical Journal. 2022 Apr 30.

●Hagde et al. Therapeutic potential of quercetin in diabetic foot ulcer: Mechanistic insight, challenges, nanotechnology driven strategies and future prospects. Journal of Drug Delivery Science and Technology. 2022 Jul 8:103575. 13;27(8):2498.

Comment 3) The first paragraph of the introduction (Cited to refs 1-4) is too general and should be deleted. A strong background is required to convince the reader why updating a review about quercetin is required.

Comment 4) The amounts of quercetin reported in Figure 1 needs an appropriate citation.

Comment 5) Page 3 section 2: The authors say “More than twenty plant species…… contain quercetin.” This claim needs a reference. More than 20 can be hundreds to thousands. What are the main plant families containing quercetin? Is there a phylogeny relation through evolution between the named plants containing quercetin in this section?

Comment 6) The authors attributed various biological effects to quercetin. These effects need appropriate discussion to compare equivalent concentrations to prevent side effects. For example, on page 8, section 3.3, the authors say that quercetin has anti-S. aureus effect at a dose of 20 mg/ml. Then, what are the ID50 values for quercetin antitumor effects? I mean beneficial pharmacological effects only have the chance to be translated clinically if they do not have detrimental side effects such as cytotoxicity against normal cells. 

Comment 7) Figure 3 is too general and is not acceptable at this level of science. Separated mechanistic figures with details of the biological processes displaying the interference of quercetin should be drawn. Similarly, other sections, such as section 3.5 about anti-inflamatory mechanisms, and the subsequent sections need professional figures.

Comment 8) A new column should be added to Table 1 to define if the experiments are performed in vitro, ex vivo, or in vivo in animal models or in clinical trials.

Comment 9) The scientific names of plant, bacterial, and fungal species must be in italics throughout the manuscript, such as on Page 3, section 2:

Santalum album, Mangifera indica, Emblica ofcinalis, Curcuma domestica valenton, Withania somnifera, Foeniculum vulgare, and Cuscuta refexa

Comment 10) The conclusion is a series of general effects and needs to be rewritten.

Moderate English modifications are required.

Author Response

The authors decided to write a narrative review on the health benefits of quercetin according to the title. This manuscript does not address a new specific gap in the field since similar titles have been updated recently. There are fundamental problems in the bibliography, as mentioned in the comments. The recently published materials on this title are ignored by the authors, which is totally against publication ethics. Besides declaring the requirement of a new submission (despite the existence of similar titles from 2020-2022) the authors need to work hard on the literature survey, appropriate figures, and integrated discussion to organize the manuscript. Otherwise, the manuscript cannot be further considered in its current status.

We would like to thank you for the time and effort taken to review our manuscript, we have carefully responded to each comment. Your requested corrections carried out in this second draft are in track changes.

 Comment 1) The authors should define the new aspects of their manuscript compared to similar articles published recently and the reason for their ignorance. These are only some examples:

  • Septembre-Malaterre et al. Focus on the high therapeutic potentials of quercetin and its derivatives. Phytomedicine Plus. 2022 Jan 15:100220.
  • Deepika, Maurya PK. Health benefits of quercetin in age-related diseases. Molecules. 2022 Apr
  • Salehi et al. Therapeutic potential of quercetin: New insights and perspectives for human health. Acs Omega. 2020 May 14;5(20):11849-72.

In the articles that you mentioned, for example, in article 1, only three perspectives of quercetin, such as antioxidant, antiviral, and anti-inflammatory, have been discussed.

In the second article, a short review of antioxidant, anti-inflammatory, antidiabetic, and age-associated diseases has been discussed. Also, the authors were satisfied with brief explanations about these characteristics.

And in the third article, only the anti-diabetic, anti-Alzheimer, and anti-parasitic effects are mentioned.

In this article, we covered a wide range of quercetin uses in pharmaceuticals and health benefits, as well as the most important characteristics of quercetin, such as antioxidant, anticancer, antibacterial, anti-Alzheimer, antiviral, antifungal activity, anti-inflammatory, anti-obesity, anti-diabetes, antihypertensive, antiallergic, anti-asthmatic, and the bioavailability of quercetin.

Comment 2) There are lots of published papers on quercetin therapeutic potentials with appropriate inclusion criteria, which have been neglected without a known reason. What were the keywords, databases, range of years etc… to consider an article? The authors should limit their investigation to the recent three years and update their manuscript for neglected data. These are only some examples:

  • Nutmakul T. A review on benefits of quercetin in hyperuricemia and gouty arthritis. Saudi Pharmaceutical Journal. 2022 Apr 30.
  • Hagde et al. Therapeutic potential of quercetin in diabetic foot ulcer: Mechanistic insight, challenges, nanotechnology driven strategies and future prospects. Journal of Drug Delivery Science and Technology. 2022 Jul 8:103575. 13;27(8):2498.

According to your suggestion, these two articles, plus more recent articles, have been added to the manuscript.

In order to write this article, we used the most relevant keywords to our work, such as antimicrobial, antioxidant, anticancer, antifungal, anti-inflammatory, bioavailability of quercetin, etc. Moreover, all data was obtained from valid sources. To write this manuscript, 90% of the articles from the last 5 years from Scopus, ScienceDirect, and Google Scholar have been used.

 Comment 3) The first paragraph of the introduction (Cited to refs 1-4) is too general and should be deleted. A strong background is required to convince the reader why updating a review about quercetin is required.

The paragraph mentioned has been deleted and replaced.

 Comment 4) The amounts of quercetin reported in Figure 1 needs an appropriate citation.

The citation has been added to the caption of the figure.

 Comment 5) Page 3 section 2: The authors say “More than twenty plant species…… contain quercetin.” This claim needs a reference. More than 20 can be hundreds to thousands. What are the main plant families containing quercetin? Is there a phylogeny relation through evolution between the named plants containing quercetin in this section?

The reference to the sentence has been added to the manuscript. In the main article, the authors mentioned that “Quercetin is found in many fruits, vegetables and in more than twenty species of plants particularly in Mangifera indica, Emblica officinalis, Withania somnifera, Cuscuta reflexa, Santalum album, Curcuma domestica valenton and Foeniculum vulgare.”

According to the phylogeny, we did not find any relation between the mentioned plants in the manuscript containing quercetin.

 Comment 6) The authors attributed various biological effects to quercetin. These effects need appropriate discussion to compare equivalent concentrations to prevent side effects. For example, on page 8, section 3.3, the authors say that quercetin has anti-S. aureus effect at a dose of 20 mg/ml. Then, what are the ID50 values for quercetin antitumor effects? I mean beneficial pharmacological effects only have the chance to be translated clinically if they do not have detrimental side effects such as cytotoxicity against normal cells.

The effective dosage for a microorganism is different in research studies. The effective dosage of quercetin differs depending on its health beneficial properties. As a result, there is no effective dosage to determine and report for these side effects. Hence, more studies are needed to determine the specific dosage of quercetin and the side effects caused by it in the future.

Comment 7) Figure 3 is too general and is not acceptable at this level of science. Separated mechanistic figures with details of the biological processes displaying the interference of quercetin should be drawn. Similarly, other sections, such as section 3.5 about anti-inflamatory mechanisms, and the subsequent sections need professional figures.

The figure 3 has been completely changed and the new figures has been added to the other sections.

Comment 8) A new column should be added to Table 1 to define if the experiments are performed in vitro, ex vivo, or in vivo in animal models or in clinical trials.

The desired column has been added along with the requested information.

 Comment 9) The scientific names of plant, bacterial, and fungal species must be in italics throughout the manuscript, such as on Page 3, section 2:

Santalum album, Mangifera indica, Emblica ofcinalis, Curcuma domestica valenton, Withania somnifera, Foeniculum vulgare, and Cuscuta refexa

All the scientific names were changed to italics.

 Comment 10) The conclusion is a series of general effects and needs to be rewritten.

The conclusion has been totally changed, and its body was strengthened by adding future perspective.

Round 2

Reviewer 2 Report

The response to most comments is not accurate. It is revised in a careless style without enough concern.

The response to comment 7 is not convincing. Visual communication is vital to research advancement. It means that traditional, text-heavy communication methods are inadequate. The modification in figure 3 is not enough. Several separated figures with details are required for pharmacological effects. For sure it needs more time and work.

 When the databases and the range of years are not added to the main text of the manuscript the reader assumes that some serendipitous papers are gathered together. The authors should define how did they reach to a total of 214 references inside the manuscript. I just mentioned some few examples of the references that were ignored by the authors in the first round of the review. There might be several other citations that are ignored by the authors.

Moderate English language modification is required.

Author Response

The response to most comments is not accurate. It is revised in a careless style without enough concern.

We would like to thank you for the time and effort taken to review our manuscript, we have carefully responded to each comment. Your requested corrections carried out in this second draft are in track changes.

The response to comment 7 is not convincing. Visual communication is vital to research advancement. It means that traditional, text-heavy communication methods are inadequate. The modification in figure 3 is not enough. Several separated figures with details are required for pharmacological effects. For sure it needs more time and work.

Figure 3 has been modified and and the most important signalling pathways affected by quercetin during cancer prevention and each mechanism have been added in this figure.

A new figure (figure 4) as the anti-inflammation signal pathways of quercetin has been added to indicate the details of anti-inflammation mechanism of quercetin. 

 When the databases and the range of years are not added to the main text of the manuscript the reader assumes that some serendipitous papers are gathered together. The authors should define how did they reach to a total of 214 references inside the manuscript. I just mentioned some few examples of the references that were ignored by the authors in the first round of the review. There might be several other citations that are ignored by the authors.

A new section (review methodology) has been added to explain the citation method.

Round 3

Reviewer 2 Report

None